

# Urine proteome changes in rats subcutaneously inoculated with approximately ten tumor cells

Jing Wei, Wenshu Meng and Youhe Gao

Department of Biochemistry and Molecular Biology, Beijing Normal University, Gene Engineering Drug and Biotechnology Beijing Key Laboratory, Beijing, China

## ABSTRACT

**Background**. Biomarkers are changes associated with the disease. Urine is not subject to homeostatic control and therefore accumulates very early changes, making it an ideal biomarker source. Usually, we have performed urinary biomarker studies involving at least thousands of tumor cells. However, no tumor starts from a thousand tumor cells. We therefore examined urine proteome changes in rats subcutaneously inoculated with approximately ten tumor cells.

**Methods**. Here, we serially diluted Walker-256 carcinosarcoma cells to a concentration of $10^2$/mL and subcutaneously inoculated 0.1 mL of these cells into nine rats. The urine proteomes on days 0, 13 and 21 were analyzed by liquid chromatography coupled with tandem mass spectrometry.

**Results**. Hierarchical clustering analysis showed that the urine proteome of each sample at three time points were clustered into three clusters, indicating the good consistency of these nine rats when inoculated with the same limited tumor cells. Differential proteins on days 13 and 21 were mainly associated with cell adhesion, autophagic cell death, changes in extracellular matrix organization, angiogenesis, and the pentose phosphate pathway. All of these enriched functional processes were reported to contribute to tumor progression and could not be enriched through random allocation analysis.

**Conclusions**. Our results indicated that (1) the urine proteome reflects changes associated with cancer even with only approximately ten tumor cells in the body and that (2) the urine proteome reflects pathophysiological changes in the body with extremely high sensitivity and provides potential for a very early screening process of clinical patients.

## INTRODUCTION

Urine is an ideal biomarker resource. Blood often remains stable because of homeostatic mechanisms. However, as the filtrate of blood, urine has no need to remain stable and thus tolerates a much higher degree of changes. Therefore, urine can accumulate all changes from the whole body and may provide the potential to detect early and small changes in the body (*Gao, 2013*). Urine is easily affected by various physiological factors, such as sex, age and diet (*Wu & Gao, 2015*). In patients, the urine proteome is easily influenced by certain medications because of necessary therapeutic measures. Therefore, our laboratory

Corresponding author
Youhe Gao, gaoyouhe@bnu.edu.cn

proposed a strategy for urinary biomarker studies. First, we used animal models to find early biomarkers of related diseases. Then, we verified candidate biomarkers in clinical urine samples (*Gao, 2014*). The use of animal models minimizes external influencing factors, such as diet, gender, age, medications and some environmental factors. In addition, using animal models will allow identification of the exact start of the disease, which is helpful in the early detection of cancer. Differential urinary proteins found in animal models are likely to be directly associated with related diseases. According to this strategy, our laboratory has applied different types of animal models, such as subcutaneous tumor-bearing model (*Wu, Guo & Gao, 2017a*), pulmonary fibrosis model (*Wu et al., 2017b*), glioma model (*Ni et al., 2018*), liver fibrosis model (*Zhang et al., 2018a*), Alzheimer's disease model (*Zhang et al., 2018b*), chronic pancreatitis model (*Zhang, Li & Gao, 2018c*) and myocarditis model (*Zhao et al., 2018*), to search for early biomarkers before pathological changes and clinical manifestations.

Urine can reflect changes more sensitively than blood. It has been reported that even when interference is introduced into the blood with two anticoagulants, changes in the abundance of more proteins were consistently detected in urine samples than in plasma (*Li, Zhao & Gao, 2014*). In addition, the urine proteome has been applied to detect tumors in various tumor-bearing animals. For example, (i) in W256 subcutaneously tumor-bearing rats, a total of ten differential urinary proteins were identified before a tumor mass was palpable (*Wu, Guo & Gao, 2017a*); (ii) in the intracerebral W256 tumor model, nine urinary proteins changed significantly before any obvious clinical manifestations or abnormal magnetic resonance imaging (MRI) signals (*Zhang et al., 2019*); (iii) in the glioma rat model, a total of thirty differential proteins were identified before MRI (*Ni et al., 2018a*); (iv) a total of seven urinary proteins changed in both lung tumor-bearing mice and lung cancer patients, indicating their potential roles in the early detection of lung cancer (*Zhang et al., 2015*); (v) in a urothelial carcinoma rat model, differential urinary proteins from upregulated biological processes might be seen as candidate biomarkers (*Ferreira et al., 2015*). All of these studies were performed involving thousands of tumor cells; however, no tumor starts from a thousand tumor cells. Since urine is a more sensitive biomarker resource than blood, we explored the sensitivity limit of urine. We determined whether the urine proteome changes if there are only a small number of tumor cells in the body.

In this study, we subcutaneously injected approximately ten Walker-256 carcinosarcoma cells into nine rats. Urine samples were collected on days 0, 13, and 21. Urine proteins were analyzed by liquid chromatography-tandem mass spectrometry (LC-MS/MS). Differential proteins on days 13 and 21 were analyzed by functional enrichment analysis to find associations with tumor progression. This research aimed to determine whether the urine proteome could reflect changes associated with these ten tumor cells. The technical flowchart is presented in Fig. 1.

## MATERIAL AND METHODS

### Animal treatment

Male Wistar rats ($n = 12$, $150 \pm 20$ g) were purchased from Beijing Vital River Laboratory Animal Technology Co., Ltd. Animals were maintained with a standard laboratory diet
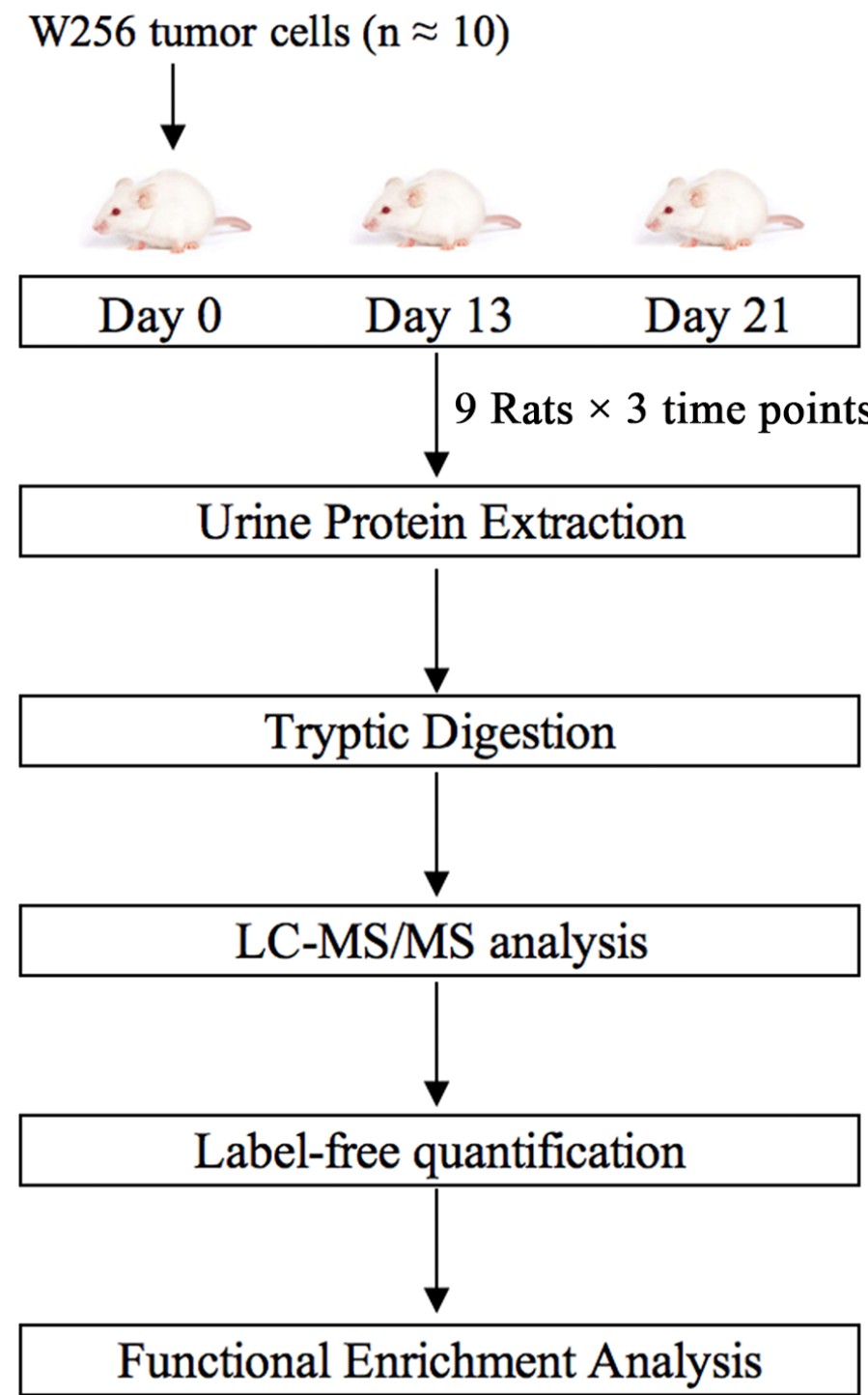

**Figure 1 Workflow of protein identification in rats subcutaneously inoculated with ten tumor cells.**
Urine was collected on days 0, 13 and 21 after inoculation with tumor cells. Urinary proteins were extracted, digested, and identified by liquid chromatography coupled with tandem mass spectrometry (LC-MS/MS). Functional enrichment analysis of differential proteins was performed by DAVID and IPA.

under controlled indoor temperature ($21 \pm 2\,°C$), humidity (65–70%) and 12 h/12 h light–dark cycle conditions. The experiment was approved by Peking Union Medical College (Approval ID: ACUC-A02-2014-008).

Walker-256 (W256) carcinosarcoma cells were purchased from the Cell Culture Center of the Chinese Academy of Medical Sciences (Beijing, China). W256 tumor cells were intraperitoneally inoculated into Wistar rats. The W256 ascites tumor cells were harvested from the peritoneal cavity after seven days. After two cell passages, the W256 ascites tumor cells were collected, centrifuged, and resuspended in 0.9% normal saline (NS). Then, W256 tumor cells were serially diluted to a concentration of $10^2$/mL. The viability of W256 cells was evaluated by the Trypan blue exclusion test using a Neubauer chamber, and only 95% viable tumor cells were used for serially dilution.

The rats were randomly divided into the following two groups: rats subcutaneously inoculated with tumor cells ($n = 9$) and control rats ($n = 3$). In the experimental group, rats were inoculated with 10 W256 cells in 100 µL of NS into the right flank of the rats. The control rats were subcutaneously inoculated with an equal volume of NS. All rats were anesthetized with sodium pentobarbital solution (four mg/kg) before inoculation.

### Urine collection

Before urine collection, all rats were accommodated in metabolic cages for 2–3 days. Urine samples were collected from rats subcutaneously inoculated with tumor cells ($n = 9$) on days 0, 13 and 21. All rats were placed in metabolic cages individually for 12 h to collect urine without any treatment. After collection, urine samples were stored immediately at $-80\,°C$.

### Extraction and digestion of urinary proteins

Urine samples ($n = 27$) were centrifuged at $12,000 \times g$ for 30 min at $4\,°C$. Then, the supernatants were precipitated with three times the volume of ethanol at $-20\,°C$ overnight. The pellets were dissolved sufficiently in lysis buffer (8 mol/L urea, 2 mol/L thiourea, 50 mmol/L Tris, and 25 mmol/L DTT). After centrifugation at $4\,°C$ and $12,000 \times g$ for 30 min, the protein samples were measured by using the Bradford assay. A total of 100 µg of each protein sample was digested with trypsin (Trypsin Gold, Mass Spec Grade, Promega, Fitchburg, WI, USA) by using filter-aided sample preparation (FASP) methods (*Wisniewski et al., 2009*). These digested peptides were desalted using Oasis HLB cartridges (Waters, Milford, MA, USA) and then dried by vacuum evaporation (Thermo Fisher Scientific, Bremen, Germany).

### LC-MS/MS analysis

Digested peptides ($n = 27$) were dissolved in 0.1% formic acid to a concentration of 0.5 µg/µL. For analysis, one µg of peptide from each sample was loaded into a trap column (75 µm ×2 cm, three µm, C18, 100 Å) at a flow rate of 0.25 µL/min and then separated with a reversed-phase analytical column (75 µm ×250 mm, two µm, C18, 100 Å). Peptides were eluted with a gradient extending from 4%–35% buffer B (0.1% formic acid in 80% acetonitrile) for 90 min and then analyzed with an Orbitrap Fusion Lumos Tribrid Mass Spectrometer (Thermo Fisher Scientific, Waltham, MA, USA). The MS data were acquired

using the following parameters: (i) data-dependent MS/MS scans per full scan were auquired at the top-speed mode; (ii) MS scans had a resolution of 120,000, and MS/MS scans had a resolution of 30,000 in Orbitrap; (iii) HCD collision energy was set to 30%; (iv) dynamic exclusion was set to 30 s; (v) the charge-state screening was set to +2 to +7; and (vi) the maximum injection time was 45 ms. Each peptide sample was analyzed twice.

## Label-free quantification

Raw data files ($n = 54$) were searched using Mascot software (version 2.5.1; Matrix Science, London, UK) against the Swiss-Prot rat database (released in February 2017, containing 7,992 sequences). The parent ion tolerance was set to 10 ppm, and the fragment ion mass tolerance was set to 0.02 Da. The carbamidomethylation of cysteine was set as a fixed modification, and the oxidation of methionine was considered a variable modification. Two missed trypsin cleavage sites were allowed, and the specificity of trypsin digestion was set for cleavage after lysine or arginine. Dat files ($n = 54$) were exported from Mascot software and then processed using Scaffold software (version 4.7.5, Proteome Software Inc., Portland, OR). The parameters were set as follows: both peptide and protein identifications were accepted at a false discovery rate (FDR) of less than 1.0% and proteins were identified with at least two unique peptides. Different samples were compared after normalization with the total spectra. Protein abundances at different time points were compared with spectral counting, according to previously described procedures (*Old et al., 2005*; *Schmidt et al., 2014*).

## Statistical analysis

Average normalized spectral counts of each sample were used for the following statistical analysis. The levels of proteins identified on days 13 and 21 were compared with their levels on day 0. Differential proteins were selected with the following criteria: unique peptides $\geq 2$; fold change $\geq 1.5$ or $\leq 0.67$; average spectral count in the high-abundance group $\geq 3$; comparison between two groups were conducted using two-sided, unpaired $t$-test; and $P$-values of group differences were adjusted by the Benjamini and Hochberg method (*Benjamini & Hochberg, 1995*). Group differences resulting in adjusted $P$-values <0.05 were considered statistically significant. All results are expressed as the mean $\pm$ standard deviation.

## Functional enrichment analysis

Differential proteins on days 13 and 21 were analyzed by Gene Ontology (GO) based on the biological process, cellular component and molecular function categories using the Database for Annotation, Visualization and Integrated Discovery (DAVID) (*Huang da, Sherman & Lempicki, 2009*). The biological pathway enrichment at the two time points was analyzed with IPA software (Ingenuity Systems, Mountain View, CA, USA).

# RESULTS

## Characterization of rats subcutaneously inoculated with tumor cells

A total of 12 male Wistar rats (150 $\pm$ 20 g) were randomly divided into the following two groups: a control group ($n = 3$) and a group of rats subcutaneously inoculated with W256

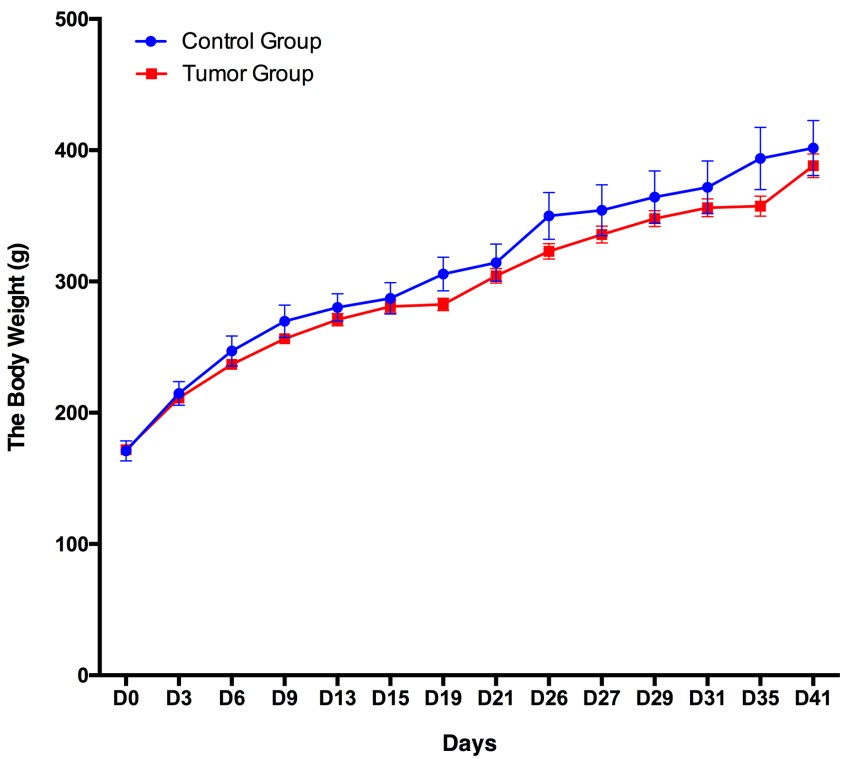

**Figure 2** Body weight changes between the rats subcutaneously inoculated with tumor cells and the control rats.

tumor cells ($n = 9$). The body weight of these 12 rats was recorded every 3–5 days, and the daily behavior changes of the two groups were observed. The body weight of the group of rats subcutaneously inoculated with W256 tumor cells was slightly lower than that of the rats in the control group, but there were no significant differences until day 41 (Fig. 2). In addition, we did not observe any detectable tumor mass in the whole period. The rats in the control group performed normal daily activities and had shiny hair. There were no significant differences in daily behavior between these two groups.

## Urine proteome changes

Twenty-seven urine samples at three time points (days 0, 13, and 21) were used for label-free LC-MS/MS quantitation. A total of 824 urinary proteins with at least 2 unique peptides were identified with <1% FDR at the protein level (Table S1). A hierarchical clustering was performed by using the complete linkage method. As shown in Fig. 3A, all technical replications within one sample were clustered together, indicating that the technical variation was smaller than the interindividual variation. In addition, all 824 proteins were clustered into three clusters, which almost corresponded to the urine proteome samples from the same group on day 0, day 13 and day 21 (except rat4-D0 and rat6-D21), indicating that intragroup technical variation was smaller than the intergroup biological variation and showed good consistency among these nine rats when inoculated with the same

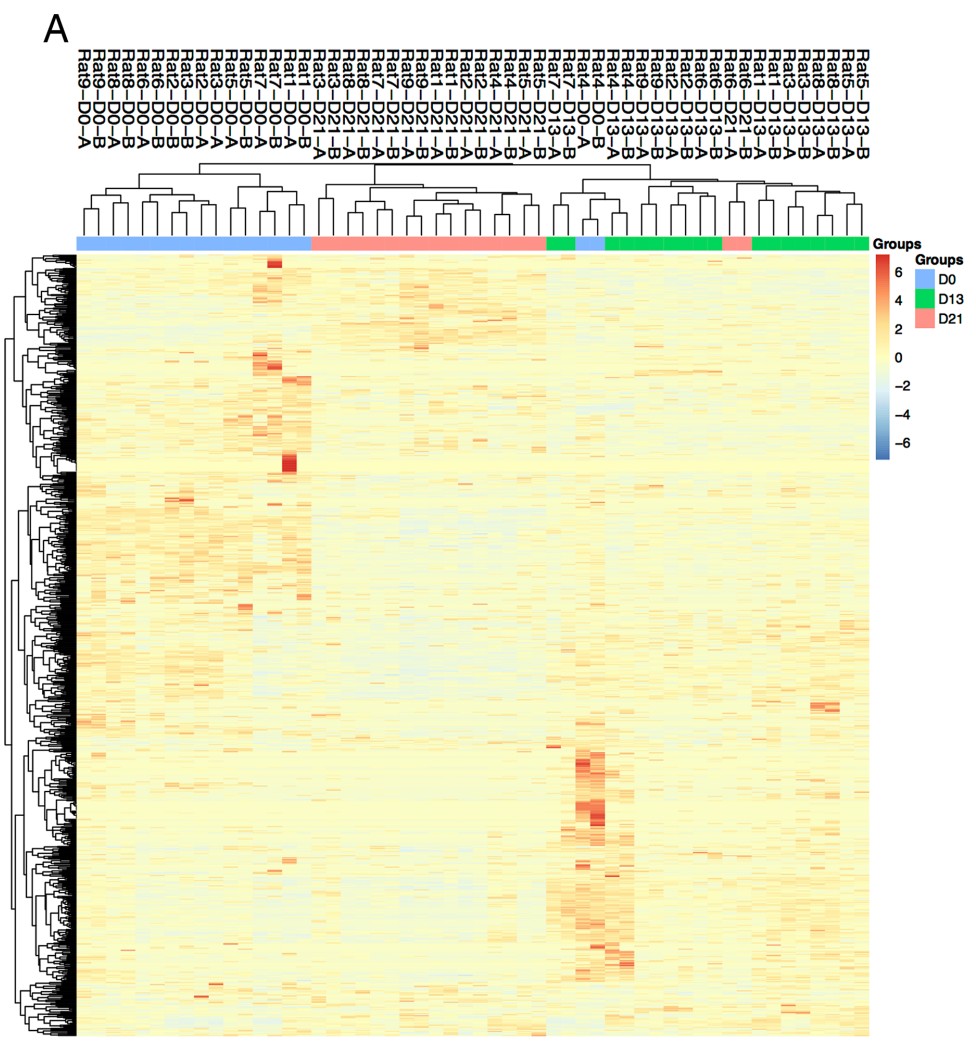

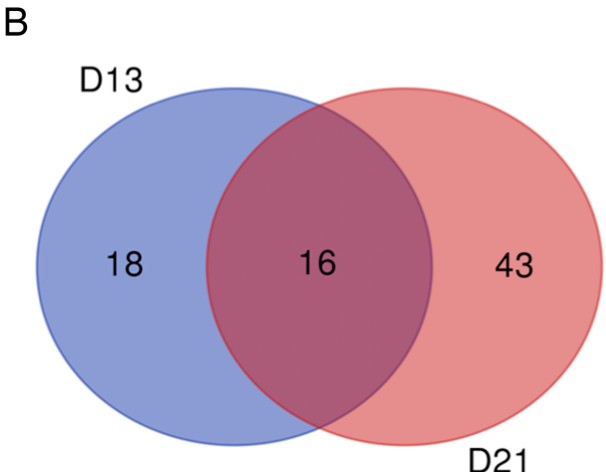

**Figure 3** **Proteomic analysis of urine samples on days 13 and 21 in rats subcutaneously inoculated with tumor cells.** (A) Unsupervised cluster analysis of all proteins identified by LC-MS/MS. (B) Overlap evaluation of the differential proteins identified on days 13 and 21.

limited number of tumor cells. Using screening criteria, 34 and 59 differential proteins were identified on days 13 and 21, respectively. The overlap of these differential proteins is shown by a Venn diagram in Fig. 3B. Details are presented in Table 1.

## Functional analysis

Functional enrichment analysis of differential proteins was performed by DAVID (*Huang da, Sherman & Lempicki, 2009*). Differential proteins were classified into biological processes, cellular components and molecular functions. The major biological pathways of differential proteins were enriched by IPA software. A significance threshold of $P < 0.05$ was used in all these representative lists.

Lists of fourteen representative biological processes on days 13 and 21 are presented in Fig. 4A. Cell adhesion, negative regulation of endopeptidase activity and organ regeneration were overrepresented both on days 13 and 21. Blood coagulation, acute-phase response, autophagic cell death, positive regulation of cell proliferation, extracellular matrix organization, and response to glucose were independently overrepresented on day 13. On day 21, heterophilic cell–cell adhesion via plasma membrane cell adhesion molecules, proteolysis, positive regulation of phagocytosis and angiogenesis were independently enriched.

To identify the biological pathways involved with the differential urine proteins, IPA software was used for canonical pathway enrichment analysis. A total of 18 and 13 significant pathways were enriched on days 13 and 21, respectively (Fig. 4B). Among these pathways, enriched intrinsic prothrombin activation pathway, coagulation system, acute phase response signaling, extrinsic prothrombin activation pathway and NAD phosphorylation and dephosphorylation were overrepresented on days 13 and 21. In addition, some representative pathways, such as autophagy, phagosome maturation and the role of tissue factor in cancer, were independently enriched on day 13, and the complement system was enriched only on day 21.

The enriched cellular components and molecular functions are presented in Fig. S1. The majority of differential proteins were derived from the extracellular exosomes, extracellular space and extracellular matrix (Fig. S1A). In the molecular function category, serine-type endopeptidase inhibitor activity and calcium ion binding were overrepresented on both on days 13 and 21 (Fig. S1B).

## Random allocation statistical analysis

To confirm that these differential proteins on days 13 and 21 were indeed due to the ten subcutaneously inoculated W256 tumor cells, we randomly allocated the data of these 27 samples (Number 1 to Number 27) into three groups. We tried three random allocations, and the numbers in these three groups are shown in Table 2. In each iteration, we used the data of group 1 as the control group. When we used the previous criteria to screen differential urinary proteins, it was found that the adjusted $P$-values value on days 13 and 21 were all >0.05. No differential proteins were selected in these three randomly allocated trials. Details are shown in Tables S2, S3 and S4.

**Table 1** Differentially proteins identified on day 13 and day 21.

| Uniprot ID | Human ortholog | Description | Trends | ANOVA *P*-value | Average fold change | |
|---|---|---|---|---|---|---|
| | | | | | Day 13 | Day 21 |
| P02761 | NO | Cluster of Major urinary protein | ↑ | <0.00010 | 4.70 | 7.59 |
| P06760 | P08236 | Beta-glucuronidase | ↑ | 0.0073 | 3.56 | 4.55 |
| Q9JI85 | P80303 | Nucleobindin-2 | ↑ | 0.00018 | 2.74 | 3.69 |
| P27590 | P07911 | Uromodulin | ↑ | <0.00010 | 2.31 | 2.65 |
| P97603 | Q92859 | Neogenin (Fragment) | ↓ | <0.00010 | 0.43 | 0.21 |
| P20611 | P11117 | Lysosomal acid phosphatase | ↓ | <0.00010 | 0.56 | 0.46 |
| Q9ESS6 | P50895 | Basal cell adhesion molecule | ↓ | <0.00010 | 0.56 | 0.40 |
| Q63556 | NO | Serine protease inhibitor A3M (Fragment) | ↓ | <0.00010 | 0.47 | 0.48 |
| O35112 | Q13740 | CD166 antigen | ↓ | <0.00010 | 0.41 | 0.33 |
| Q63416 | Q06033 | Inter-alpha-trypsin inhibitor heavy chain H3 | ↓ | <0.00010 | 0.32 | 0.23 |
| O70535 | P42702 | Leukemia inhibitory factor receptor | ↓ | <0.00010 | 0.29 | 0.35 |
| P13596 | P13591 | Neural cell adhesion molecule 1 | ↓ | <0.00010 | 0.28 | 0.36 |
| P35444 | P49747 | Cartilage oligomeric matrix protein | ↓ | <0.00010 | 0.27 | 0.42 |
| P07897 | P16112 | Aggrecan core protein | ↓ | <0.00010 | 0.23 | 0.30 |
| P26453 | P35613 | Basigin | ↓ | 0.00094 | 0.23 | 0.21 |
| Q9EPF2 | P43121 | Cell surface glycoprotein MUC18 | ↓ | <0.00010 | 0.18 | 0.08 |
| P38918 | O95154 | Aflatoxin B1 aldehyde reductase member 3 | ↑ | 0.0028 | 3.36 | – |
| Q63530 | Q96BW5 | Phosphotriesterase-related protein | ↑ | 0.0004 | 3.18 | – |
| P24268 | P07339 | Cathepsin D | ↑ | <0.00010 | 1.56 | – |
| P18292 | P00734 | Prothrombin | ↓ | <0.00010 | 0.60 | – |
| P04937 | P02751 | Fibronectin | ↓ | <0.00010 | 0.61 | – |
| P08592 | NO | Amyloid beta A4 protein | ↓ | <0.00010 | 0.56 | – |
| Q91XN4 | Q13145 | BMP and activin membrane-bound inhibitor homolog | ↓ | 0.003 | 0.55 | – |
| Q9JLS4 | Q6FHJ7 | Secreted frizzled-related protein 4 | ↓ | 0.00053 | 0.52 | – |
| Q63467 | P04155 | Trefoil factor 1 | ↓ | 0.00026 | 0.50 | – |
| Q62930 | P02748 | Complement component C9 | ↓ | 0.0055 | 0.49 | – |
| P24090 | P02765 | Alpha-2-HS-glycoprotein | ↓ | 0.00026 | 0.47 | – |
| P97546 | Q9Y639 | Neuroplastin | ↓ | <0.00010 | 0.47 | – |
| P26644 | P02749 | Beta-2-glycoprotein 1 | ↓ | 0.00016 | 0.46 | – |
| Q5HZW5 | Q9NPF0 | CD320 antigen | ↓ | <0.00010 | 0.45 | – |
| Q8JZQ0 | P09603 | Macrophage colony-stimulating factor 1 | ↓ | 0.00013 | 0.42 | – |
| P07154 | P07711 | Cathepsin L1 | ↓ | 0.00021 | 0.40 | – |
| D3ZTE0 | P00748 | Coagulation factor XII | ↓ | <0.00010 | 0.25 | – |
| P07171 | P05937 | Cluster of Calbindin | ↓ | <0.00010 | 0.05 | – |
| Q63475 | Q92932 | Receptor-type tyrosine-protein phosphatase N2 | ↓ | <0.00010 | – | 6.25 |
| P00714 | P00709 | Alpha-lactalbumin | ↑ | <0.00010 | – | 3.77 |
| Q05175 | P80723 | Brain acid soluble protein 1 | ↑ | <0.00010 | – | 3.69 |
| Q4V885 | Q5KU26 | Collectin-12 | ↑ | 0.0011 | – | 2.28 |
| O55004 | P34096 | Ribonuclease 4 | ↑ | 0.00023 | – | 2.13 |

**Table 1** (*continued*)

| Uniprot ID | Human ortholog | Description | Trends | ANOVA *P*-value | Average fold change | |
|---|---|---|---|---|---|---|
| | | | | | Day 13 | Day 21 |
| P81828 | NO | Urinary protein 2 | ↑ | <0.00010 | – | 1.96 |
| P62986 | P62987 | Ubiquitin-60S ribosomal protein L40 | ↑ | 0.0011 | – | 1.81 |
| P36374 | NO | Prostatic glandular kallikrein-6 | ↑ | 0.0034 | – | 1.78 |
| P80202 | P36896 | Activin receptor type-1B | ↑ | <0.00010 | – | 1.76 |
| P06866 | P00739 | Haptoglobin | ↑ | 0.00036 | – | 1.68 |
| P07151 | P61769 | Beta-2-microglobulin | ↑ | <0.00010 | – | 1.65 |
| P08937 | NO | Odorant-binding protein | ↑ | 0.002 | – | 1.51 |
| P05544 | NO | Serine protease inhibitor A3L | ↓ | 0.00026 | – | 0.64 |
| P29598 | P00749 | Urokinase-type plasminogen activator | ↓ | <0.00010 | – | 0.62 |
| P05545 | NO | Serine protease inhibitor A3K | ↓ | <0.00010 | – | 0.61 |
| Q01460 | Q01459 | Di-N-acetylchitobiase | ↓ | <0.00010 | – | 0.61 |
| Q642A7 | Q8WW52 | Protein FAM151A | ↓ | 0.00045 | – | 0.58 |
| P21704 | P24855 | Deoxyribonuclease-1 | ↓ | <0.00010 | – | 0.56 |
| P63018 | P11142 | Cluster of Heat shock cognate 71 kDa protein | ↓ | 0.00045 | – | 0.56 |
| P61972 | P61970 | Nuclear transport factor 2 | ↓ | 0.0024 | – | 0.54 |
| Q68FQ2 | Q9BX67 | Junctional adhesion molecule C | ↓ | 0.0011 | – | 0.53 |
| P31211 | P08185 | Corticosteroid-binding globulin | ↓ | 0.00011 | – | 0.48 |
| Q9R0T4 | P12830 | Cadherin-1 | ↓ | <0.00010 | – | 0.47 |
| Q562C9 | Q9BV57 | 1,2-dihydroxy-3-keto-5-methylthiopentene dioxygenase | ↓ | 0.0057 | – | 0.43 |
| O88917 | NO | Adhesion G protein-coupled receptor L1 | ↓ | <0.00010 | – | 0.42 |
| Q9WUW3 | P05156 | Complement factor I | ↓ | <0.00010 | – | 0.42 |
| P01026 | P01024 | Complement C3 | ↓ | 0.0071 | – | 0.42 |
| P85971 | O95336 | 6-phosphogluconolactonase | ↓ | 0.0005 | – | 0.39 |
| Q63751 | NO | Vomeromodulin (Fragment) | ↓ | 0.012 | – | 0.31 |
| Q9EQV6 | O14773 | Tripeptidyl-peptidase 1 | ↓ | 0.0033 | – | 0.30 |
| P53813 | P07225 | Vitamin K-dependent protein S | ↓ | 0.00096 | – | 0.29 |
| Q00657 | Q6UVK1 | Chondroitin sulfate proteoglycan 4 | ↓ | <0.00010 | – | 0.28 |
| P04073 | P20142 | Gastricsin | ↓ | 0.00044 | – | 0.27 |
| O70244 | O60494 | Cubilin | ↓ | <0.00010 | – | 0.25 |
| Q1WIM1 | Q8NFZ8 | Cell adhesion molecule 4 | ↓ | <0.00010 | – | 0.24 |
| Q5I0D5 | Q9H008 | Phospholysine phosphohistidine inorganic pyrophosphate phosphatase | ↓ | <0.00010 | – | 0.20 |
| Q63678 | P25311 | Zinc-alpha-2-glycoprotein | ↓ | <0.00010 | – | 0.20 |
| Q99PW3 | Q99519 | Sialidase-1 | ↓ | 0.00025 | – | 0.18 |
| Q641Z7 | Q92484 | Acid sphingomyelinase-like phosphodiesterase 3a | ↓ | <0.00010 | – | 0.18 |
| P04785 | P07237 | Protein disulfide-isomerase | ↓ | <0.00010 | – | 0.17 |
| Q6AYE5 | Q86UD1 | Out at first protein homolog | ↓ | <0.00010 | – | 0.16 |
| P47820 | P12821 | Angiotensin-converting enzyme | ↓ | 0.0016 | – | 0.13 |
| Q80WD1 | Q86UN3 | Reticulon-4 receptor-like 2 | ↓ | 0.017 | – | 0.04 |

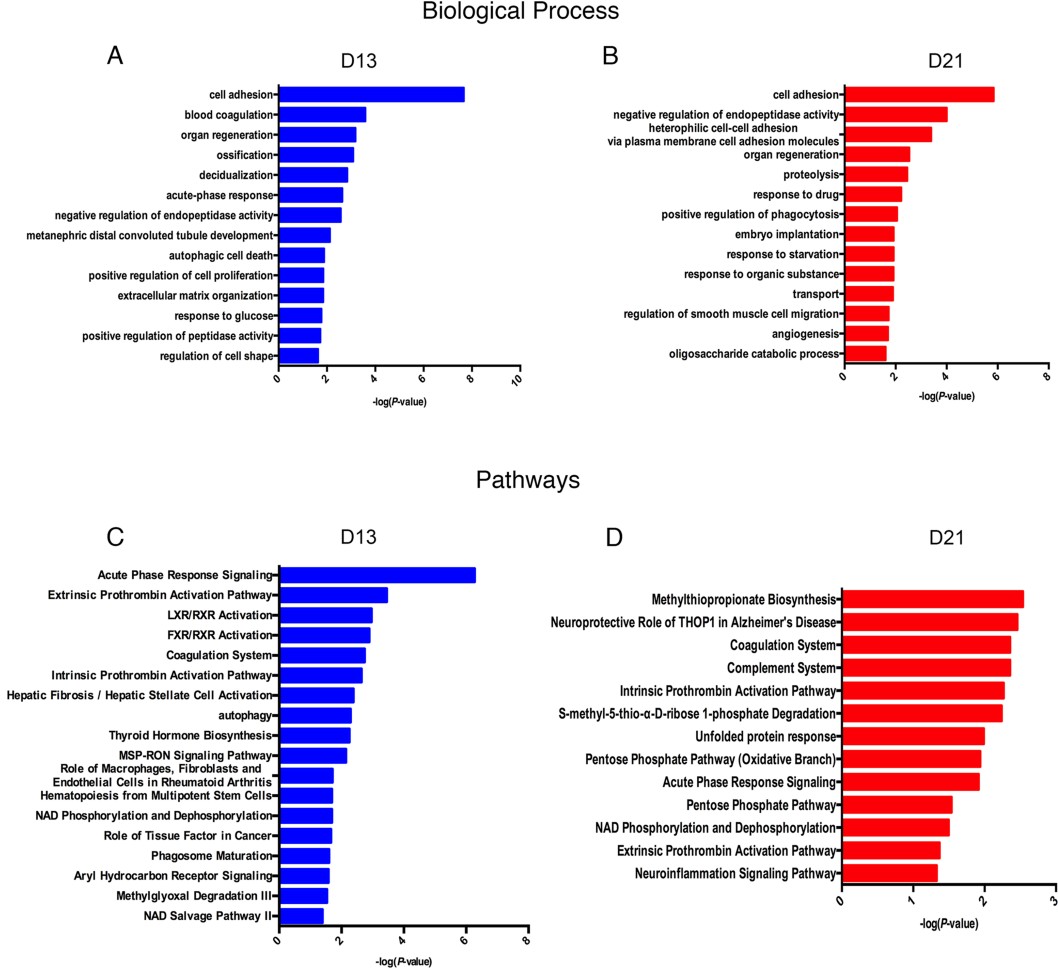

**Figure 4  Functional analysis of differential proteins on days 13 and 21.** (A) Dynamic changes in biological processes on day 13. (B) Dynamic changes in biological processes on day 21. (C) Dynamic changes in pathways on day 13. (D) Dynamic changes in pathways on day 21.

**Table 2  Random allocation of the twenty-seven urine samples.**

| Randomly allocated | Group 1 | Group 2 | Group 3 | Adjusted *P*-value |
|---|---|---|---|---|
| 1 | 1, 4, 5, 12, 13, 14, 19, 20, 21 | 2, 3, 7, 11, 15, 17, 18, 22, 23 | 6, 8, 9, 10, 16, 24, 25, 26, 27 | NO |
| 2 | 1, 2, 3, 12, 13, 15, 25, 26, 27 | 4, 5, 6, 10, 11, 14, 19, 20, 21 | 7, 8, 9, 16, 17, 18, 22, 23, 24 | NO |
| 3 | 1, 7, 9, 11, 13, 16, 19, 21, 22 | 3, 4, 5, 10, 17, 18, 20, 25, 26 | 2, 6, 8, 12, 14, 15, 23, 24, 27 | NO |

**Notes.**
Numbers 1–9 represent Rat 1-D0 to Rat 9-D0; Numbers 10–18 represent Rat 1-D13 to Rat 9-D13; Numbers 19–27 represent Rat 1-D21 to Rat 9-D21.

# DISCUSSION

Urine is an early and sensitive biomarker source that has been used for the early detection of cancer either in both animal models or clinical patients (*Beretov et al., 2015*; *Wu, Guo & Gao, 2017a*). However, no tumor starts from thousands of tumor cells. In this study, we subcutaneously inoculated approximately ten tumor cells into each of nine rats.

Unsupervised clustering analysis showed the good consistency after inoculation. A total of 34 and 59 differential proteins identified on days 13 and 21, respectively, and no urinary proteins changed after random allocation analysis.

After the functional enrichment analysis, we found that some enriched biological processes were reported to be associated with tumor progression. For example, (i) cell adhesion was usually reported to show a reduced number of tumor cells since 1962 (*Holmberg, 1962*); (ii) autophagic cell death occurs via the activation of autophagy, which has been reported to play roles in tumor suppression (*Mathew et al., 2009*); (iii) the positive regulation of cell proliferation is a common characteristic of cancer, and the inhibition of cancer cell proliferation may serve as a potential target for cancer treatment (*He et al., 2018*); (iv) changes in extracellular matrix organization were reported with crucial roles in cancer metastasis (*Goreczny et al., 2017*; *Sada et al., 2016*); (v) positive regulation of blood coagulation was frequently reported in cancer progression (*Tikhomirova et al., 2016*); and (vi) angiogenesis is still considered a common characteristic of tumorigenesis in many studies (*Baltrunaite et al., 2017*; *Protopsaltis et al., 2019*; *Ziegler et al., 2016*).

In addition, we found that some pathways were reported to play important roles in cancer. For example, (i) autophagy was reported to inhibit tumor progression (*Peng et al., 2016*); (ii) the MSP-RON signaling pathway was reported to play important roles in epithelial tumorigenesis (*Ma et al., 2010*) and will facilitate metastasis in prostate cancer cells (*Yin et al., 2017*); (iii) tissue factor (TF) expressed by tumor cells was reported to facilitate lung tumor progression (*Han et al., 2017*); (iv) upregulation of the pentose phosphate pathway (PPP) has been reported in several types of cancer (*Rao et al., 2015*); and (v) the enriched complement system pathway was reported to enhance the metastatic process of ovarian cancer cells (*Cho et al., 2016*). Our results indicated that even when limited tumor cells are present in the body, the urine proteome can reflect changes associated with cancer.

When comparing differential proteins identified in our research to W256 subcutaneously tumor-bearing model (*Wu, Guo & Gao, 2017a*) and intracerebral W256 tumor model (*Zhang et al., 2019*), we found that the proportion of overlapping proteins was small, and more than half of the differential proteins in each of the three tumor models were unique (Fig. S2). Comparing the differential proteins of the model inoculated with ten W256 cells and the W256 subcutaneously tumor-bearing model showed 25 overlapping proteins. We hypothesized that these small overlapping proteins may be due to the very different numbers of W256 tumor cells in these two animal models. Upon comparing the differences in the urine proteome between the ten tumor cell inoculated model and the intracerebral W256 tumor model, only 16 differential proteins overlapped, indicating that differential proteins were very different when the same tumor cells existed in different body parts. We also suppose that these proteomic profiles were different because the changes observed in this study were related to the very early phase of the tumor. Despite the small proportion of overlapping proteins, we found that cell adhesion was enriched in GO biological process analysis using either the 25 or 16 common differential proteins. This reduction in cell adhesion is a common characteristic of tumor cells (*Cavallaro & Christofori, 2001*). These

results suggest that although the tumor cell number and location differ, using limited tumor cells has the potential to simulate the early phase of tumor development.

Notably, it was difficult to ensure that exactly ten tumor cells were subcutaneously inoculated into each of nine rats. Given the limited number of animals in this preliminary study, a larger number of animals should be considered in future studies.

## CONCLUSIONS

In this study, we aimed to observe changes in the urine proteome when inoculating approximately ten tumor cells into nine rats. Our results indicated that (1) the urine proteome reflects changes associated with cancer, even with a limited number of tumor cells in the body, and (2) the urine proteome reflects pathophysiological changes in the body with extremely high sensitivity, providing the potential for a very early screening process in clinical patients.

### Funding

This work was supported by the National Key Research and Development Program of China (2018YFC0910202 and 2016YFC1306300), the Beijing Natural Science Foundation (7172076), the Beijing Cooperative Construction Project (110651103), the Beijing Normal University (11100704), and the Peking Union Medical College Hospital (2016-2.27). The funders had no role in study design, data collection and analysis, decision to publish, or preparation of the manuscript.

### Grant Disclosures

The following grant information was disclosed by the authors:
National Key Research and Development Program of China: 2018YFC0910202, 2016YFC1306300.
Beijing Natural Science Foundation: 7172076.
Beijing Cooperative Construction Project: 110651103.
Beijing Normal University: 11100704.
Union Medical College Hospital: 2016-2.27.

### Competing Interests

The authors declare there are no competing interests.

### Author Contributions

- Jing Wei conceived and designed the experiments, performed the experiments, analyzed the data, contributed reagents/materials/analysis tools, prepared figures and/or tables, authored or reviewed drafts of the paper, approved the final draft.
- Wenshu Meng performed the experiments, contributed reagents/materials/analysis tools.
- Youhe Gao conceived and designed the experiments, contributed reagents/materials/-analysis tools, authored or reviewed drafts of the paper, approved the final draft.

## Animal Ethics

The following information was supplied relating to ethical approvals (i.e., approving body and any reference numbers):

The experiment was approved by Peking Union Medical College (Approval ID: ACUC-A02-2014-008).

## Data Availability

The data is available at figshare: Wei, Jing; Gao, Youhe (2019): Urine proteome changes in rats subcutaneously inoculated with approximately ten tumor cells. figshare. Dataset. https://figshare.com/articles/Urine_proteome_changes_in_rats_subcutaneously_inoculated_with_approximately_ten_tumor_cells_/9221084.

## Supplemental Information

Supplemental information for this article can be found online at http://dx.doi.org/10.7717/peerj.7717#supplemental-information.

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
