# Peer review of "Urine proteome changes in rats subcutaneously inoculated with approximately ten tumor cells"

_PeerJ, doi:10.7717/peerj.7717_

## Round 0.1 · original submission · Major Revisions

Dear authors,

Please, I recommend to carefully revise the paper according to the comments of the referees. I recommend also to further clarify the manuscript and support it by other studies rather than those from your group.

[]

Reviewer 1 ·

Basic reporting

Although the hypothesis for the present study is well defined in the Introduction section, authors fail in giving sufficient background. Are there previous reports (excluding those from the group) that have successfully employed urine to detect changes in the proteome of tumor-bearing rodents? In this context, previous studies of the group related to other pathological states do not provide any significant background for this section. The English style is ok. Raw data has been shared.

Experimental design

Authors have employed the rat-derived Walker-256 carcinosarcoma cell line. Remarkably, authors have injected a low number of cells (~10 cells) to interrogate whether it is sufficient to cause any changes in the urine protein content. Changes in the body weight as well as in the behavior were followed for 41 days (according to figure 2). However, there is no mention of tumor growth. Do the animals develop any detectable tumor mass upon injection of 10 cells? Results clearly demonstrate that inoculation of a low number of tumor cells is sufficient to alter the protein composition of tumor cell-inoculated rats at days 13 and 21 post-inoculum. The proteomic analysis performed here is fine. However, in order to attribute the proteomic changes to tumor development, it would be essential to repeat the assays employing non-viable tumor cells.

Validity of the findings

As state above, the results presented in this study should be clearly correlated with the presence of viable, proliferating tumor cells. Remarkably, the discussion section largely fails in comparing the proteomic profiles herein found to previous studies that employed urine form tumor-bearing animal models or patients. Rather, the authors discussed the possible roles of proteins/pathways identified. Consequently, most of the cited articles are review studies.

Reviewer 2 ·

Basic reporting

Paper has tons of english language issues. When the english is ignored the idea of the paper mentioned is to analyze the effect of small amount of cancerous cells on the proteomic/biomarker profile of the urine and using this proteomic profile to detect cancer.

Abstract:

The abstract in not well written, the abstract background is not clear and needs to be rewritten. The abstract results has grammatical errors "were almost clustered together".

Introduction:

line 64-66: rephrase the sentence and grammar
line 68: rephrase "therapeutic measures for patients are inevitable"
line 72: please mention what makes up external influence factors
line 77: rephrase and check grammar of "appearance of pathology changes"
line 79-81: rephrase these two lines as they are not clear
line 83: check grammar
line 86: Please dont use questions, rephrase with passive voice

Materials and Methods:
line 121: rephrase "three volumes of ..."

Results:
line 190: rephrase and check grammar of "All identified urinary proteins were performed with unsupervised clustering analysis
line 190: "almost clustered together" makes very little scientific sense, mention why these are close or similar with distances between centroids or something

Discussion:
line 236: incorrect usage: "stars"
line 238: incorrect usage of "protein changed"
line 239: use "respectively"

line 240-276: needs complete rewrite, discussion feels rushed. Each paragraph is written as one complete sentence with numbers. I would suggest the authors to create separate paragraphs for each important result obtained and elaborate on each of these results. There are sufficient references, but how the results obtained here match or corroborate with previously reported studies needs connection.

line 277: incorrect usage "be avoided"
line 289: incorrect usage "helped do"

Experimental design

The materials and methods are descriptive enough for replication and has sufficient references for various sources.

The experiments are well planned and the inference of the obtained data is valid. The authors should elaborate on each findings of the differential proteomic biomarker identified in this paper. It is great that the results match the hypothesis, but a little more explanation on why these proteomic profile differ needs expanded on.

Validity of the findings

I understand that the mice are expensive to carry out these experiments, but the number of mice that yielded these results are low and the authors have mentioned this limitation in line 278-279.

Additional comments

Please use a english language editor to check for grammar and spelling mistakes.

---

## Round 0.2 · accepted · Accept

The current version of the paper is accepted for publication in PeerJ.